# Single-Top Quark Production at CMS [†]

**Priyanka \*, Kirti Ranjan, Ashutosh Bhardwaj and on behalf of CMS Collaboration**

Centre for Detector and Related Software Technology, Department of Physics and Astrophysics,
University of Delhi, Delhi 110007, India; Kirti.Ranjan@cern.ch (K.R.); ashutosh.bhardwaj@cern.ch (A.B.)

\* Correspondence: priyanka.priyanka@cern.ch; Tel.: +91-1112-7667-036

† This paper is based on the talk at the 7th International Conference on New Frontiers in Physics (ICNFP 2018), Crete, Greece, 4–12 July 2018.

**Abstract:** An overview of recent results of single-top quark production at the LHC using data collected with the CMS detector is presented. The CMS experiment has measured the electroweak production of the top quark in three production modes, namely t-channel, tW-channel, and s-channel. Measurements of the rare processes involving a single-top quark with a Z boson and a single-top quark with a $\gamma$ are also discussed. All measurements are in agreement with the standard model prediction, and no sign of physics beyond the standard model is observed.

**Keywords:** CMS; top physics; proton-proton collisions

## 1. Introduction

Ever since the discovery of the top quark in pair production mode ($t\bar{t}$) via strong interaction by the CDF and the D0 collaborations at the Tevatron in 1995 [1,2], the study of the top quark has assumed great interest. The top quark is the heaviest elementary particle, and because of its short lifetime, it decays before hadronizing, pre-dominantly into a W boson and a b-quark. Electroweak production of the top quark, which is known as single-top quark production, was also observed by the CDF and D0 collaborations at the Tevatron in 2009 [3,4]. Top quark studies, thus, provide important tests of the two main interactions in the Standard Model (SM). It is also expected to play an important role in physics Beyond the Standard Model (BSM). The cross-section of the electroweak production of top quark provides a direct probe for the $V_{tb}$ matrix element of the Cabibbo–Kobayashi–Maskawa (CKM) matrix, which may get enhanced by the non-SM couplings of the $W_{tb}$ vertex. Single-top quark production is a background for the precision $t\bar{t}$ physics and for many other SM and BSM searches, and it can be used to constrain the Parton Distribution Functions (PDFs). The single-top quark is produced via three subprocesses. The most dominant process is the t-channel in which a top quark is produced along with a light flavored quark or with a light flavored quark and a b-quark. The second most dominant process at the LHC is the production of a top quark in association with a W boson, which is known as the tW-channel. The least dominant process at the LHC is the s-channel, in which a top quark is produced along with a b-quark. Figure 1 shows the leading order Feynman diagrams for the various single-top quark production modes. The CMS at the LHC is able to probe rare processes such as the associated production of a single-top quark with a Z boson and a single-top quark in association with a $\gamma$ in t-channel mode due to the increase in center-of-mass energy and luminosity.

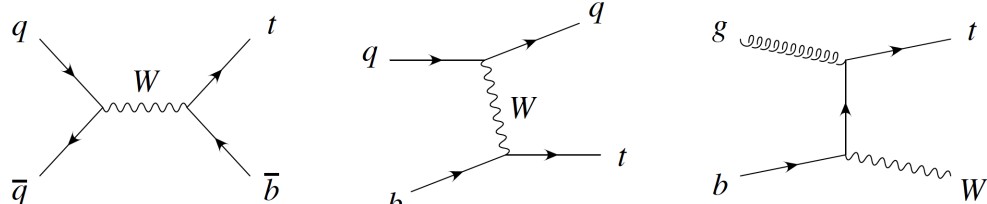

**Figure 1.** Leading order Feynman diagrams for the single-top quark production (only one contribution is shown here) in (**left**) s-channel subprocess, (**middle**) t-channel subprocess, and (**right**) tW subprocess.

## 2. t-Channel

The t-channel is the dominant mode of single-top quark production contributing around 73% of the total single-top quark production at the LHC. It provides a precise probe of the $|V_{tb}|$ matrix element of the CKM matrix. New physics beyond the SM would alter the couplings and affect the polarization: the distribution of the angle between the recoil jet in the rest-frame of the top quark and the charged lepton from the top decay is directly related to the spin asymmetry and therefore to the polarization. The t-channel is also suited for testing the proton parton distribution functions and comparing the various theoretical models with data. The cross-sections for the production of single-top quark and antiquark, in the t-channel, along with its ratio, are measured at $\sqrt{s}$ = 13 TeV using full 2016 data, corresponding to an integrated luminosity of 35.9 fb$^{-1}$ at the CMS experiment [5]. The ATLAS experiment also performed t-channel studies at 13 TeV [6]. The t-channel signature consists of a single-top quark produced in association with a recoiling light quark. The leptonic decay of the top quark is considered, and the events with one lepton, either electron or muon, and two jets are selected, where one of the two jets originates from a bottom quark (b-tag). Measurements were performed by the simultaneous maximum likelihood fit on multivariate discriminators, separately for the signal region described above and two control regions. which are 3 jets with 1 b-tag and 3 jets with 2 b-tags and for the different lepton flavors and the lepton charge. In total, 12 discriminator distributions were fitted simultaneously. The fit was repeated twice, once to extract the single-top quark and anti-top quark production cross-sections in which top and anti-top channel signal strengths were the free parameters and then to extract the ratio in which anti-top channel signal strength and the ratio of the two processes used as free parameters. The two-fit approach allows one to get rid of the covariance matrix. The main systematic uncertainty was the signal modeling, which covers the potential mismodeling of the t-channel signal process, renormalization and factorization scale uncertainty, matching of matrix element and parton shower, parton shower scale etc.

Figure 2 shows the measured ratio of the cross-sections of the top and anti-top processes, $R_{t-ch} = \frac{\sigma(t)}{\sigma(\bar{t})}$, which was measured to be $1.65 \pm 0.02$ (*stat.*) $\pm 0.04$ (*syst.*). It is compared to the recent predictions using different PDFs to describe the inner structure of the proton. A good agreement with most of the PDF sets was found within the uncertainties of the measurement. Furthermore, with the experimental uncertainty on the ratio, getting close to the theoretical uncertainties of the PDF models, the data will be able to constrain the PDF. The total uncertainty on the measured ratio is about two-times the size of the uncertainty in the predictions from theory. The total cross-section for the production of the t-channel is measured to be $219 \pm 1.5$ (*stat.*) $\pm 32.9$ (*syst.*)pb. From the ratio of the measured cross-section to the theoretical cross-section, the absolute value of the CKM matrix element was measured to be $f_{LV}V_{tb} = 1.01 \pm 0.05$ (*exp*) $\pm 0.02$ (*theo*), with the assumption that branching fractions to other quarks, like $V_{td}$ and $V_{ts}$, are significantly smaller than the b-quark branching fraction. $f_{LV}$ is the anomalous form factor, which takes the possible presence of an anomalous $W_{tb}$ coupling into account. $f_{LV}$ is one for the SM. All of these measurements are in agreement with the SM predictions, within the uncertainties.

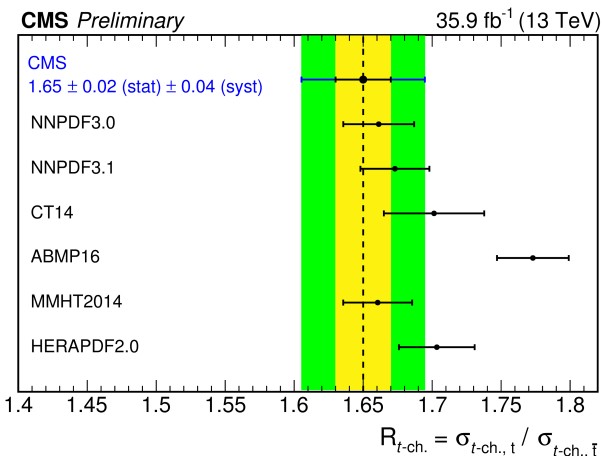

**Figure 2.** Ratio $R_{t-ch}$ is shown for the measurement and expectations from different PDFs. A top quark mass of 172.5 GeV is used as a nominal value for powheg4FScalculation. The uncertainties for different PDFs come from statistical uncertainty, factorization, and renormalization scales' uncertainties and the top quark mass uncertainties. In the case of the measurement value, the inner uncertainty band is for the statistical one and the outer uncertainty band for total uncertainty.

## 3. tW-Channel

tW-channel production contains one fourth of the total single-top quark production at the LHC. However, there exists a major challenge to observe the tW process, which is to overcome the interference with top quark pair production at next-to-leading order for the extraction of the tW signal. To overcome this theoretical difficulty in the definition of tW production, two configurations [7] were used to subtract the overlapping diagrams: the Diagram Subtraction (DS) and the Diagram Removal (DR). The DR removes the resonant $t\bar{t}$ effect at the amplitude level, whereas the DS removes the effect at the cross-section level by re-shuffling the kinematics. The difference between the two provides the size of the interference. At the CMS, we performed the first measurement of the tW production cross-section at center-of-mass energy of 13 TeV using the full 2016 data corresponding to an integrated luminosity of 35.9 fb$^{-1}$ [8]. The ATLAS experiment also performed the corresponding tW-channel inclusive studies at 13 TeV [9]. The signature of this analysis is a single-top quark produced in association with a W boson, both of which further decay leptonically, and the events with two opposite sign leptons (which are electrons or muons) and one jet (which originates from a bottom quark) have been selected. However, we defined three regions for the signal extraction, which are one jet, which is tagged as b-jet (1j1t); and two jets with either one or both being tagged as b-jets (2j1t or 2j2t). The signal strength is determined from the simultaneous maximum likelihood fit to the Boosted Decision Tree (BDT) distributions in the 1j1t and 2j1t regions and the sub-leading jet $p_T$ distribution in the 2j2t region.

Figure 3 shows the data/MC comparison of the number of jets that are tagged as b-jets after the dilepton selections. The main systematic uncertainties come from the jet energy scaling, lepton identification, and $t\bar{t}$ modeling. The CMS measured the cross-section of the tW-channel as $\sigma = 63.1 \pm 1.8\ (stat.) \pm 6.4\ (syst.) \pm 2.1\ (lumi)$ pb, with a relative uncertainty of 11%, which is in agreement with the SM prediction of $\sigma(NNLO) = 71.7 \pm 1.8\ (scale) \pm 3.4\ (PDF)$ pb. This is the first measurement at the $\sqrt{s} = 13$ TeV by the CMS Collaboration through BDT analysis.

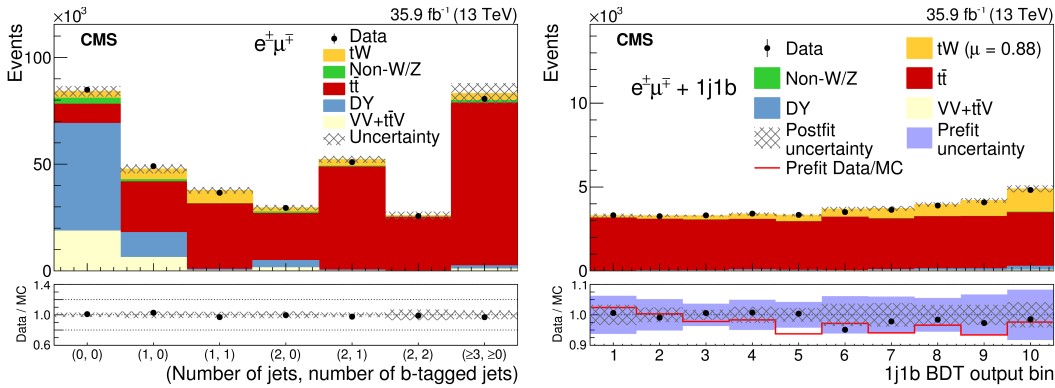

**Figure 3.** Yield comparison of observed data with the expected simulation as a function of the number of jets and the number of b-tagged jets, for the events passed dilepton selections. The uncertainty band here includes all the systematic uncertainties excluding background normalization and the statistical uncertainty. The bottom panel shows the ratio of the data to expected yields sum (**left**). Comparison of the Boosted Decision Tree (BDT) output in the one-jet one-b-tag region after the observed data and the simulated events fit is performed. The uncertainty band in the plot includes the statistical and systematic uncertainties. The bottom panel shows the data for the prediction ratios from simulations and from the fit (**right**).

## 4. s-Channel

The s-channel is a very challenging channel as it contains only 3% of the total single-top quark production at the LHC. It also grows much slower with center-of-mass energy as compared to the other single-top production modes. However, this channel is sensitive to new physics, such as searches for $W'$ and charged Higgs. Evidence for the s-channel production came from the D0 experiment [10], while observation of the same came after combining the results from both D0 and CDF collaborations. The ATLAS collaboration independently confirmed the evidence for the s-process at LHC [11]. CMS presented a search for the single-top quark production in the s-channel at $\sqrt{s} = 7$ and 8 TeV, corresponding to the integrated luminosities of 5.1 and 19.7 fb$^{-1}$, respectively [12]. The signature of this analysis is a single-top quark produced associated with a bottom quark; the top further decays leptonically, and the events with one lepton (muon channel only for $\sqrt{s} = 7$ TeV, whereas both muon and electrons for $\sqrt{s} = 8$ TeV) and two jets originating from a bottom quark are selected. For the signal extraction, a binned likelihood fit was performed on the BDT output in two regions, the two-jets two-tagged region, which is the real signal region; on the three-jets two-tagged $t\bar{t}$ region; and on the two-jets one-tagged $tq, W + jets$ regions.

Figure 4 shows the BDT discriminant distributions in the 2j2t region for the muon channel at 7 TeV and at 8 TeV, respectively. The main systematic uncertainties here come from the jet energy scaling and the b-tagging. CMS measured the cross-section of the s-channel as $\sigma = 7.1 \pm 8.1 \, (stat + syst)$ pb at $\sqrt{s} = 7$ TeV and $\sigma = 13.4 \pm 7.3 \, (stat + syst)$ pb at $\sqrt{s} = 8$ TeV with $\sigma(7 \, TeV) = 4.56 \pm 0.07 \, (scale) \pm 0.17 \, (PDF)$ pb and $\sigma (8 \, TeV) = 5.55 \pm 0.08 \, (scale) \pm 0.21 \, (PDF)$ pb being expected cross-sections, respectively. The observed (expected) signal significance of combined measurement is 2.5 standard deviations (1.1 standard deviations). The measurements agrees with the prediction of SM.

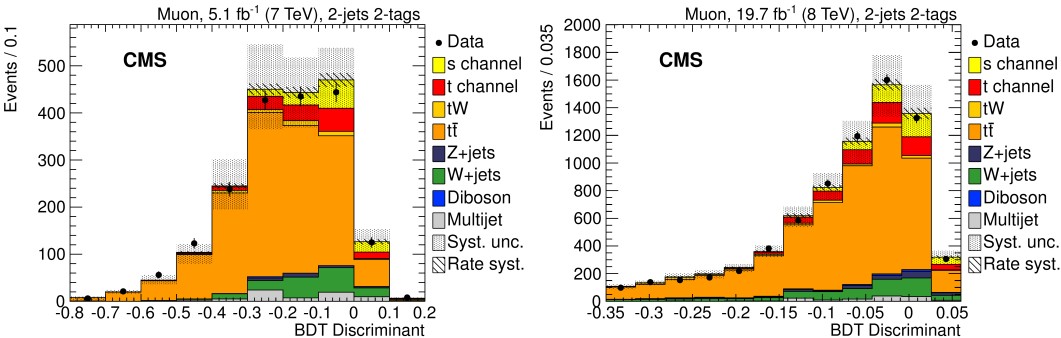

**Figure 4.** Data simulation comparison for distributions of the BDT discriminants in the two-jets two-tags region for the muon channel at $\sqrt{s} = 7$ TeV (**left**) and $\sqrt{s} = 8$ TeV (**right**). The simulation is normalized to the combined (7 + 8 TeV) fit results. The inner uncertainty bands consist only of the post-fit background rate uncertainties, and the outer ones consist of the total systematic uncertainty.

## 5. Single-Top Quark in Association with a Photon ($\gamma$), in t-Channel Mode

The large luminosity at the LHC allows searching for even rarer processes, such as the associated production of a single-top quark in the t-channel and a photon, as shown in Figure 5, known as $t\gamma$ in the following; and associated production of a single-top quark in the t-channel and a Z boson, called $tZq$ in the subsequent section. $t\gamma$ is an extremely rare process as SM radiatively generates this coupling via electroweak loop corrections, which is too small to observe at LHC. Therefore, this opens the possibility of the search for new physics in the top quark sector. Further, its cross-section is sensitive to the top quark electric charge and the top quark electric and magnetic dipole moments. CMS performed the measurement of the $t\gamma$ production cross-section at $\sqrt{s} = 13$ TeV using full data of 2016 corresponding to an integrated luminosity of 35.9 fb$^{-1}$ [13]. The signature of this analysis contains events with 1 muon, 1 photon, MET, and at least 2 jets, out of which 1 jet originates from a bottom quark. For the signal extraction, binned likelihood fit was performed on the BDT output, in the signal 1 b-tag region and control 2 b-tag ($t\bar{t}$) regions. The main systematic uncertainties come from the jet energy scaling and the signal modeling. An excess above the background-only hypothesis was observed, with a p-value of $4.27 \times 10^{-6}$, which corresponds to a significance of 4.4 standard deviation, whereas 3.0 standard deviation was expected. The CMS measured the cross-section times' branching fraction, as $\bar{B}(t \rightarrow \mu\nu b)\sigma(t\gamma j) = 115 \pm 17(stat)^{+33}_{-27}(syst)$ fb, whereas the expected standard model cross-section times' branching fraction within the central fiducial phase space is $\sigma_{t\gamma}(NLO) = 81 \pm 4 \ (scale + PDF)$ fb, which is in agreement with the measured one within the quoted uncertainties. This is the first experimental evidence of single-top quark production in association with a photon.

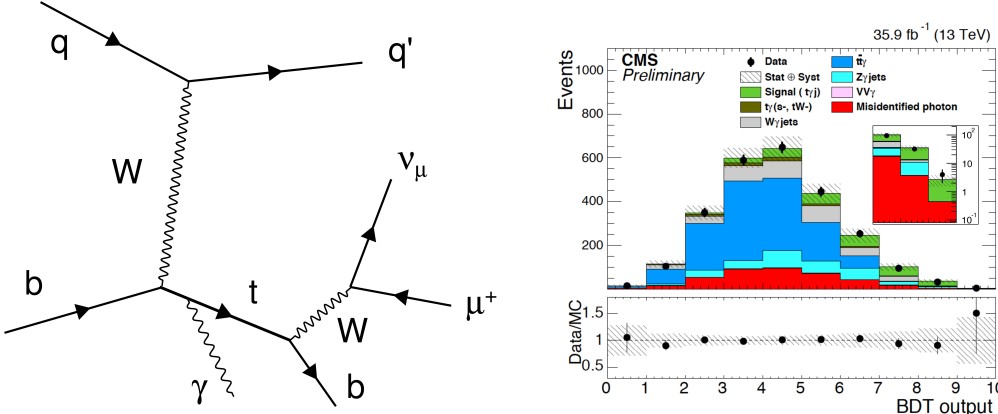

**Figure 5.** Representative Feynman diagram for single-top quark production in association with a $\gamma$ in the t-channel, including the leptonic decay of the W boson in the top quark decay (**left**). The BDT output distribution for data and SM predictions after the fit. The inset presents an enlarged portion on the last three bins (**right**).

## 6. Single-Top t-Channel with Z Emission Production

A search for the associated production of the Z boson with a single-top quark ($tZq$), for the expected SM & $tZ$ FCNC (Flavor-Changing Neutral Current) processes, has been done using full data of 2016 corresponding to an integrated luminosity of 35.9 fb$^{-1}$ at $\sqrt{s} = 13$ TeV. This very rare process is sensitive to many couplings, such as WZ, top-Z, and anomalous couplings. The signature of this analysis for both SM and FCNC final states is events with three charged leptons in the final state with a pair of the same flavor, opposite sign leptons giving an invariant mass compatible with a Z boson within a 10-GeV window, with the SM $tZ$ requiring 2-jets 1-tag and the FCNC requiring 1-jet 1-tag (single-top FCNC) or at least 2-jets and at least 1-tag ($t\bar{t}$ FCNC).

### 6.1. SM tZq

The single-top t-channel with Z emission production is an unobserved SM process involving top quarks, leading order Feynman diagrams of which are shown in Figure 6. This is also an extremely rare process of two orders of magnitude smaller than tW. As this is sensitive to ttZ and triple gauge boson Effective Field Theory (EFT) couplings, therefore, possible deviations may indicate the BSM. Main backgrounds come from ttV (where V is W and Z), WZ, and non-prompt lepton productions. Furthermore, it is the background to other SM processes such as tt + H, V and the background to the FCNC process ($tZ$). CMS measured the associated production cross-section of a single-top quark and a Z boson at the $\sqrt{s} = $ energy of 13 TeV using full data of 2016 corresponding to an integrated luminosity of 35.9 fb$^{-1}$ [14]. ATLAS performed this analysis at 36.1 fb$^{-1}$ data [15]. The $tZq$ cross-section was measured from a simultaneous binned maximum-likelihood fit to 12 distributions of BDT discriminators in the one b-tag, which is the main signal region, and two b-tag ($ttZ$) regions and to the transverse W mass distribution in the zero b-tag region for every four final states ($\mu\mu\mu, ee\mu, eee, e\mu\mu$). During the BDT training, matrix element weights were added to increase the performance, as shown in the Figure 6. CMS measured the cross-section as $\sigma(tl^+l^-q) = 123^{+33}_{-31}(stat)^{+29}_{-23}(syst)$ fb, which is compatible with the next-to- leading order SM prediction of $\sigma_{tZ(ll)q}(NLO) = 94.2 \pm 3.1$ fb within uncertainties, with an observed (expected) significance of 3.7 (3.1) standard deviations. The main systematic uncertainties come from the background normalization and the signal modeling.

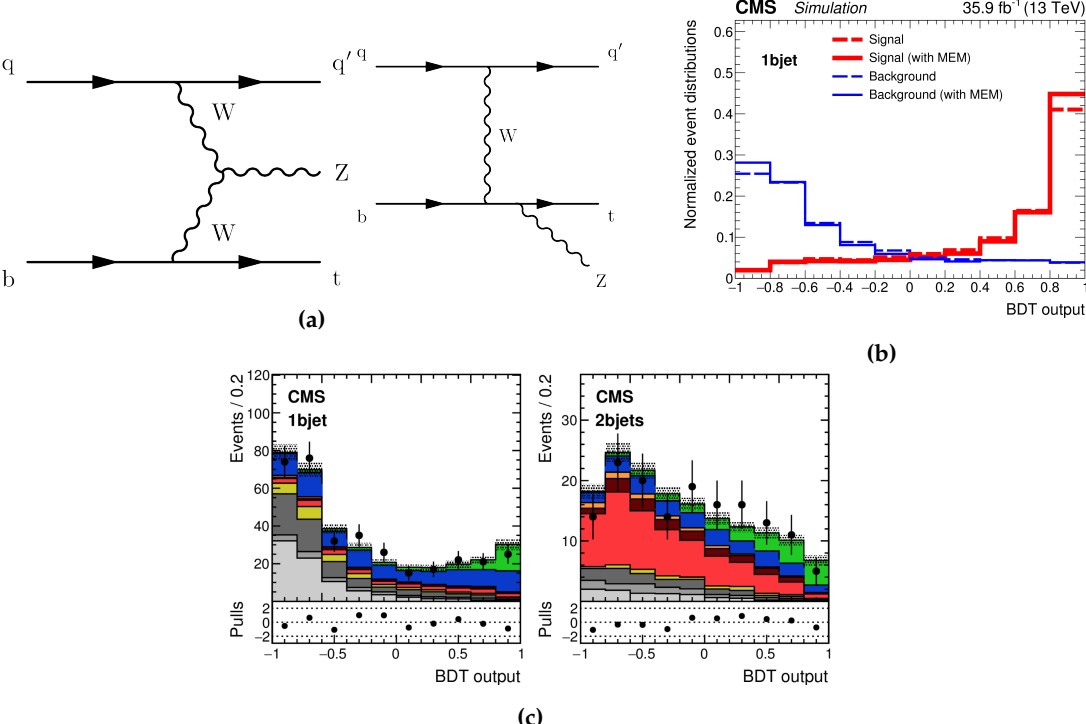

**Figure 6.** (**a**) Leading order $tZq$ production diagrams. (**b**) Thick lines represents BDT output normalized distributions for the signal, and thin lines are the representation from the simulation for the backgrounds for one-b-jet region. Dashed and solid lines represent the BDT discriminator including and excluding the MEMvariables, respectively. Signals and backgrounds include the four considered channels contributions. (**c**) Template distributions used for signal extraction: BDT discriminator in the one-b-jet region (**left**), and the BDT output in the two-b-jets control region (**right**).

*6.2. FCNC tZq*

FCNC transitions are the interaction processes where a fermion undergoes a change of flavor, as shown in Figure 7a,b, without the alteration of its charge. In SM, FCNC are forbidden at the tree level and highly suppressed at higher order. Several SM extensions enhance these FCNC interactions. CMS searched for the FCNCs involving a top quark and a Z boson at $\sqrt{s} = 13$ TeV using full data of 2016 corresponding to an integrated luminosity of 35.9 fb$^{-1}$ [16]. This search focused on single-top quark and top quark pair production FCNC interactions in the three lepton final states, where the FCNC interaction happens at the production or at the top quark decay. For this analysis, the same selections were used as the SM single-top t-channel with Z emission analysis. For the signal extraction, two simultaneous likelihood fits were performed (Higgs combine tool), one for the Single-Top Signal Region (STSR) and another one for the top quark pair signal region (TTSR) to discriminate single-top quark FCNC and top quark pair FCNC events from the backgrounds, which takes into account five different regions for the four different lepton channels ($3e, 2e1\mu, 1e2\mu, 3\mu$). The Table 1 shows those five statistically-independent different regions used in the analysis to extract limits using a likelihood fit. In the STSR, only single-top quark FCNC is used as the signal for the training, while in the TTSR, the single-top quark and top quark pair FCNC are used for training; whereas other control regions are used to constrain the non-prompt lepton backgrounds. The WZ control region focuses on Non-Prompt Leptons (NPLs) originating from Z/gamma + jets (DY+ jets) and simultaneously constrains the WZ + jets' background rate. Since the NPL rates are different for electrons and muons and the leptons not associated with the Z boson are assumed to be NPL, two different rate parameters are used. One rate is assigned to the NPL muon in the $3\mu$ and $2e1\mu$ channels, and one is assigned to the NPL electron in the 3e and 1e2$\mu$ channels. The NPL backgrounds coming from $t\bar{t} + jets$ are constrained by two control regions TTCR and STCR, one for each signal region (TTSR and STSR).

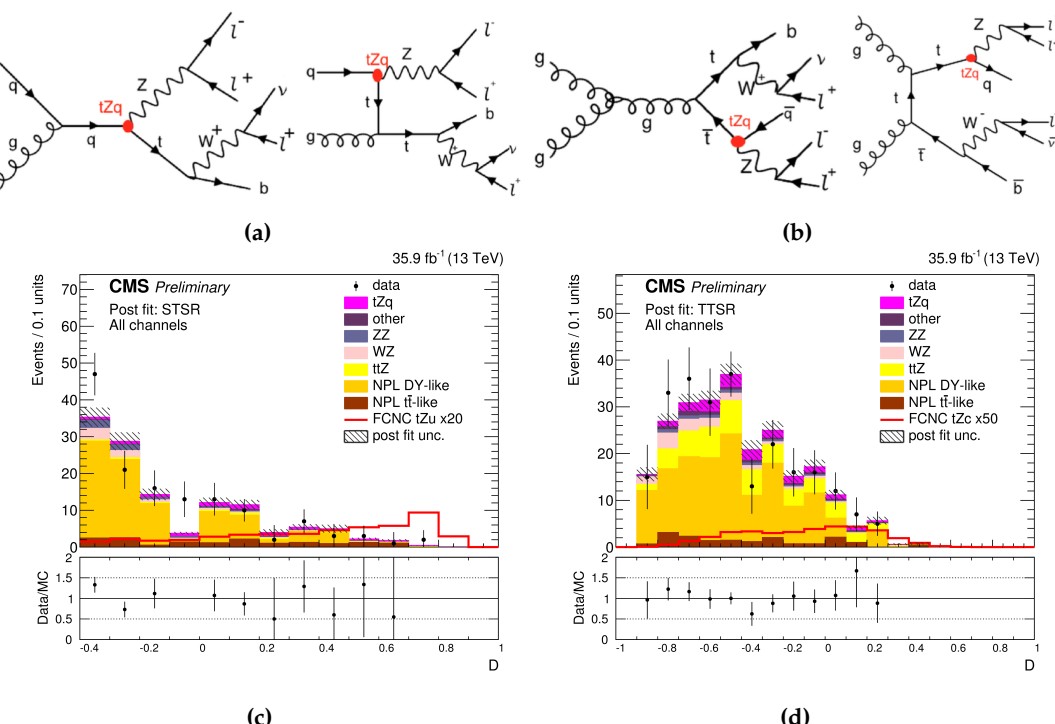

**Figure 7.** Feynman diagrams at leading order: single-top quark event (**a**) and top quark pair production (**b**). The vertex labeled $tZq$ is the sought-for Flavor-Changing Neutral Current (FCNC) interaction. The discriminating variable distribution after the fit for all different leptonic channels: single-top quark tZu (**c**) and top quark pair tZc (**d**).

**Table 1.** The statistically independent regions used in the analysis.

| | WZ Control Region (WZCR) | Single Top Quark Signal Region (STSR) | Top Quark Pair Signal Region (TTSR) | Single Top Quark Control Region (STCR) | Top Quark Pair Control Region (TTCR) |
|---|---|---|---|---|---|
| Number of jets | $\geq 1, \leq 3$ | 1 | $\geq 2, \leq 3$ | 1 | $\geq 2, \leq 3$ |
| Number of b jets | 0 | 1 | $\geq 1$ | 1 | $\geq 1$ |
| $\lvert M(Z_{\text{reco}}) - M_Z \rvert$ < 7.5 GeV | Yes | Yes | Yes | No | No |

Figure 7 shows the BDT discriminator distribution after the fit for all different leptonic channels in the single-top quark signal process and in the top quark pair production signal process for the top quark to $uZ$ and for the top quark to $cZ$ branching, respectively. The observed (expected) upper limit on the branching fraction of top quark decays to $uZ$ was found to be $B(t \to uZ) < 0.024\%$ (0.015%), and the observed (expected) upper limit on the branching fraction of the top quark decays to $cZ$ was $B(t \to cZ) < 0.045\%$ (0.037%), assuming one non-vanishing coupling at a time. The main systematic uncertainties arise from the signal modeling, jet energy scaling, and from the b-tagging, and no significant deviation was observed from the predicted background.

## 7. Discussion

This proceeding presents the CMS measurements covering a broad range of single-top quark analyses at the LHC. These analyses range from the precision measurements, i.e., inclusive cross-sections of the single-top quark in the t-channel and tW-channel have been measured using p-p collision data at $\sqrt{s} = 13$ TeV; evidence new processes, i.e., s-channel, $tZq$, and t$\gamma$; and searches for BSM processes, i.e., FCNC interaction in t-channel production. Increased luminosity during Run 2 benefited the rare single-top process searches. All presented results show good agreement with the SM.

**Author Contributions:** All authors contirbute equally to this work.

**Funding:** Priyanka acknowledges the financial support by the Department of Atomic Energy and the Department of Science and Technology, India.

**Conflicts of Interest:** The authors declare no conflict of interest. The founding sponsors had no role in the design of the study; in the collection, analyses, or interpretation of data; in the writing of the manuscript; nor in the decision to publish the results.

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
