# Peer review of "Single-Top Quark Production at CMS†"

_universe, doi:10.3390/universe5010019_

Round 1

Reviewer 1 Report

This is a nice review of single-top CMS analyses and publication will be recommended after the following mostly textual comments are addressed:

- citations have to be ordered

- L14: tevatron -> Tevatron

- L19: Single-top quark -> Single top-quark

- picobarn (pb) and femtobarn (fb) should be in roman not italics throuout the manuscript

- L90: center of mass -> center-of-mass

- L123: \times symbol should be used instead of letter "x"

- Citations to ATLAS related analyses are missing. Please amend.

Author Response

- citations have to be ordered

Done

- L14: tevatron -> Tevatron

Done

- L19: Single-top quark -> Single top-quark

Done

- picobarn (pb) and femtobarn (fb) should be in roman not italics throuout the manuscript

Done

- L90: center of mass -> center-of-mass

Done

- L123: \times symbol should be used instead of letter "x" 

Done

- Citations to ATLAS related analyses are missing. Please amend.

Done

Reviewer 2 Report

In this paper, an overview of recent results of single top quark production at the LHC, using data collected with the CMS detector, are presented. This paper is interesting and useful. I recommend its publication.

Author Response

Thanks a lot!

Reviewer 3 Report

My main concern with the paper is two-fold: First, apart from conference proceedings (which this does not appear to be) I have not seen a paper from CMS with three authors "on behalf of" the CMS Collaboration. Is this actually a paper sanctioned by the collaboration? The second concern is the English in the introduction. It is in very bad need of editing by a native English speaker. The results in the rest of the paper are OK, if unexciting.

A few specific comments by line #:

L11 - The first citations in the paper are #s 7 and 8. The first six references do not appear to be cited.

L15-16 "It is also expected to play an important role in electroweak symmetry breaking" This is speculation, not fact.

Figure 2 The green and yellow bands appear redundant with the CMS data point. Would be better to remove the data point to distinguish the measurement from the theory points.

L60 is this t+tbar?

L193 "I acknowledge..." Which one of the three authors is "I"?

Author Response

My main concern with the paper is two-fold: First, apart from conference proceedings (which this does not appear to be) I have not seen a paper from CMS with three authors "on behalf of" the CMS Collaboration. Is this actually a paper sanctioned by the collaboration? 

This is a conference proceeding based on the talk presented in "7th International Conference on New Frontiers in Physics (ICNFP2018)" where alongwith the name of author, her Supervisors' names are also listed.

The second concern is the English in the introduction. It is in very bad need of editing by a native English speaker. The results in the rest of the paper are OK, if unexciting.

Done with the English check/change in the introduction. Kindly let us know if you want any further change/comments.

A few specific comments by line #:

L11 - The first citations in the paper are #s 7 and 8. The first six references do not appear to be cited.

Done "Citations are ordered now".

L15-16 "It is also expected to play an important role in electroweak symmetry breaking" This is speculation, not fact.

Sentence has been modified to â€œ It is also expected to play an important role in Beyond Standard Model (BSM) physics"

Figure 2 The green and yellow bands appear redundant with the CMS data point. Would be better to remove the data point to distinguish the measurement from the theory points.

It is taken as it is from the CMS public results of the "CMS-PAS-TOP-17-011".

L60 is this t+tbar?

It is total t+tBar cross-section in t-channel production.

L193 "I acknowledge..." Which one of the three authors is "I"?

Sentences has been modified to "Priyanka acknowledges the financial support by the Department of Atomic Energy and the Department of Science and Technology, India."
